# Observational study of the clinical performance of a public-private partnership national referral hospital network in Lesotho: Do improvements last over time?

**Nancy A. Scott**[1]⊙*, **Jeanette L. Kaiser**[1]⊙, **Brian W. Jack**[2]‡, **Elizabeth L. Nkabane–Nkholongo**[3,4]‡, **Allison Juntunen**[1]‡, **Tshema Nash**[1]‡, **Mayowa Alade**[1]‡, **Taryn Vian**[5]⊙

**1** Department of Global Health, Boston University School of Public Health, Boston, MA, United States of America, **2** Department of Family Medicine, Boston Medical Center, Boston, MA, United States of America, **3** Lesotho Boston Health Alliance, Maseru, Lesotho, **4** School of Public Health, Sefako Makgatho University of Health Sciences, Pretoria, South Africa, **5** School of Nursing and Health Professions, University of San Francisco, San Francisco, CA, United States of America

⊙ These authors contributed equally to this work.
‡ These authors also contributed equally to this work.
* nscott@bu.edu

**Data Availability Statement:** Data for this analysis is available through the OpenBU repository at https://hdl.handle.net/2144/44912. The OpenBU

## Abstract

Public-private partnerships (PPP) may increase healthcare quality but lack longitudinal evidence for success. The Queen 'Mamohato Memorial Hospital (QMMH) in Lesotho is one of Africa's first healthcare PPPs. We compare data from 2012 and 2018 on capacity, utilization, quality, and outcomes to understand if early documented successes have been sustained using the same measures over time. In this observational study using administrative and clinical data, we assessed beds, admissions, average length of stay (ALOS), outpatient visits, and patient outcomes. We measured triage time and crash cart stock through direct observation in 2013 and 2020. Operational hospital beds increased from 390 to 410. Admissions decreased (-5.3%) while outpatient visits increased (3.8%). ALOS increased from 5.1 to 6.5 days. Occupancy increased from 82% to 99%; half of the wards had occupancy rates ≥90%, and Neonatal ward occupancy was 209%. The proportion of crash cart stock present (82.9% to 73.8%) and timely triage (84.0% to 27.6%) decreased. While overall mortality decreased (8.0% to 6.5%) and neonatal mortality overall decreased (18.0% to 16.3%), mortality among very low birth weight neonates increased (30.2% to 36.8%). Declines in overall hospital mortality are promising. Yet, continued high occupancy could compromise infection control and impede response to infections, such as COVID-19. High occupancy in the Neonatal ward suggests that the population need for neonatal care outpaces QMMH capacity; improvements should be addressed at the hospital and systemic levels. The increase in ALOS is acceptable for a hospital meant to take the most critical cases. The decline in crash cart stock completeness and timely triage may affect access to emergency treatment. While the partnership itself ended earlier than anticipated, our evaluation suggests that generally the hospital under the PPP was operational, providing high-level, critically needed services,

repository has data access policies and procedures consistent with NIH data sharing policies.

**Funding:** NAS received a grant from World Bank Group through The International Finance Corporation to support this work (Award 7191820; https://www.ifc.org/wps/wcm/connect/corp_ext_content/ifc_external_corporate_site/home). The funders developed the scope of work for the study but had no role in the study design, data collection and analysis, decision to publish or preparation of the manuscript.

**Competing interests:** The authors have declared that no competing interests exist.

and continued to improve patient outcomes. Quality at QMMH remained substantially higher than at the former Queen Elizabeth II hospital.

## Introduction

Hospitals play critical roles in health systems, delivering essential care for routine conditions and specialty care for complex and critical patients. In low- and middle-income countries (LMICs), which generally have fewer physicians per capita [1,2], public hospitals hold an even more prominent position. However, persistent health system challenges in LMICs coupled with limited resources, call for innovations in service delivery and funding.

A public-private partnership (PPP), a long-term, formalized cooperation between the public- and private-sectors, combines the competencies of both partners to achieve specific outcomes, allowing governments to address challenges and shift risk, while leveraging private sector financing and capacity [3,4]. Advocates promote them as a potential solution for health funding shortages due to fiscal constraints, while promising better quality of services with greater efficiency [5]. Opponents argue that this arrangement can result in inequity in access to care [6]. While hospital PPPs have been implemented with some successes in countries including Australia, the United Kingdom, Canada, Iran, and Turkey, there is mixed acceptance [5–9]. In 2014, an expert panel from the European Commission concluded there was, overall, insufficient evidence to determine if PPPs are an efficient mechanism to finance health systems of member states [10]. There is even less evidence from LMICs [5–9].

The integrated PPP model includes financing, construction, facility operation, and clinical service provision [4]. Some hospitals using this model have shown improved clinical outcomes. For example, hospitals in Finland and Spain have experienced reduced infection rates, lengths of stay, wait times for surgery, and readmission rates, among other improvements [5]. Yet, little longitudinal data exist on the clinical performance of integrated PPPs. Given the limited evidence on PPPs as an effective mechanism to improve quality and finance health systems, the Organization for Economic Cooperation and Development (OECD) has called for more measurement of PPP results [11].

In October 2011, the Queen 'Mamohato Memorial Hospital (QMMH) replaced Lesotho's 100-year-old national referral hospital, Queen Elizabeth II (QEII) in the capital city of Maseru. QMMH and its affiliated primary care clinics–collectively known as the QMMH Integrated Network (QMMH-IN)–is one of sub-Saharan Africa's first and most well-known healthcare PPPs, constructed and operated under an integrated model [12]. QMMH is operated by Ts'epong, a consortium of Netcare Hospital Group, a private South African health care provider, and several South African and Basotho-owned businesses. Ts'epong was selected as the private partner by the Government of Lesotho (GoL) through a competitive tender process.

The first goal of the partnership was to replace the aging QEII and upgrade the filter clinics to provide high quality, publicly funded care to the greater Maseru district and referral services for the country. The second goal was to engage the private sector to increase efficiency, accountability, and quality of care while maintaining the government's role as steward of the health sector and promulgator of policies and standards. Under the 18-year PPP contract, Ts'epong would operate QMMH-IN. Ts'epong would receive monthly payments from GoL for initial loan repayment and operating costs, including clinical services up to an agreed-upon annual threshold of 20,000 inpatient stays and 350,000 ambulatory visits.

Previous evaluations of QMMH-IN examined indicators on capacity, utilization, quality, and patient outcomes. A 2012 evaluation found that, compared to QEII, QMMH-IN provided

more services, increased quality of care, and produced better patient outcomes, including a 41% reduction in overall mortality during its first year [12,13]. Ts'epong implemented management improvements, including computerized information systems, hiring and supervision systems for human resources, preventive maintenance, and other backbone systems essential for accountability and achieving efficiency, quality, and patient outcomes [12–14].

Despite these successes, the operation of QMMH-IN has been controversial [15]. The partnership itself has been strained with disagreements between the public and private partners, and between shareholders within the private consortium. These have resulted in litigations, some still under arbitration in 2021 [16]. Payments from the GoL have frequently been delayed, and payments for services above the contract threshold have been withheld [13,16]. Within this context, we sought to understand if early successes seen in 2012 quality metrics persisted into 2018, approximately halfway through the 18-year contract.

## Materials and methods

### Study setting

The Kingdom of Lesotho has a population of 2.1 million and $2,824 GDP per capita [17,18]. Nineteen of the 21 hospitals are operated by the Ministry of Health (MoH) or by the Christian Health Association of Lesotho (CHAL) [19]. The MoH collects a nominal fee per outpatient visit (15 Maloti, ~USD1) and specific fees for select diagnostic tests and procedures, such as a CT scan (300 Maloti, ~USD20).

QMMH-IN consists of a new hospital, an ambulatory clinic (Gateway) located on the hospital campus, and three renovated primary care filter clinics spread throughout Maseru. The hospital includes an Intensive Care Unit (ICU), a Neonatal Intensive Care Unit (NICU), additional hospital bed capacity and specialist physicians, while the filter clinics have added diagnostics and services [12]. The hospital serves as the highest-level referral center for obstetric and neonatal complications in the country, but also receives self-referrals of women in labor. S1 Table describes wards in 2018, including changes since 2012, the hospital's first full year of operation. The new Neonatal ward caters to neonates requiring additional supportive care but not NICU-level high-dependency supportive care, while the Nursery generally houses healthy neonates born at QMMH. QMMH does not provide the following services per the PPP contract: chemotherapy and radiotherapy, most transplants, most joint replacements, dialysis for chronic renal disease, as well as multiple cosmetic or elective procedures.

Gateway Clinic allows for filtering of non-urgent cases and does not conduct deliveries. The three filter clinics offer ambulatory care and 24 beds total for short-term inpatient obstetric care. Deliveries requiring Caesarian section surgery or management of complications are referred to QMMH. The three filter clinics and Gateway Clinic generally refer cases to QMMH when needed.

In 2018, QMMH was unofficially functioning as a combined district hospital and tertiary referral hospital for the country. In 2020, plans to establish two regional referral hospitals were underway and a contract to construct a new district hospital in Maseru was signed. Though QEII had been reopened, it served as an outpatient-only center and referred specialty cases on.

### Objective and study design

We aimed to understand if QMMH-IN's quality achievements documented in 2012 persisted six years later. We used multiple methods to capture performance indicators on capacity, utilization, clinical quality, and patient outcomes, replicating 2012 methods where possible [12]. Data sources included administrative and clinical data, and direct observation. The 2018 (Timepoint 2) cross-section findings were compared to that of 2012 (Timepoint 1) to

understand sustained changes in performance. We have included ward-level data for capacity, utilization, and mortality metrics as they help assess change in and drivers of performance at QMMH-IN.

### Data collection

Using the QMMH-IN electronic reporting system, we extracted data on operational beds and utilization metrics. The human resources department provided staff figures. We collected patient outcome data from electronic monthly ward reports. Direct observations occurred in 2013 (Timepoint 1) and 2020 (Timepoint 2) to assess quality. S2 Table includes detailed indicator definitions, data sources, and a description of how each indicator was constructed. The indicators selected were appropriate at the time of the baseline assessment in 2009. Because this was not initially designed as a longitudinal study and because of the evolving context of the PPP, some indicators changed over time and other measures of quality were not included. We opted to use the same measures that were initially selected to ensure comparability over time and utility for key stakeholders.

### Measures

**Capacity.**   We measured operational beds and clinical/non-clinical staffing. The total number of operational beds was obtained per month for each ward and the three filter clinics, excluding mortuary beds and Nursery cradles. Due to changes during 2018, operational beds are reported as an average over 12 months.

**Utilization.**   We measured admissions, inpatient days, and ambulatory visits. We calculated average length of stay (ALOS) and occupancy rates. Nursery admissions of ill neonates have been included in 2012 and 2018 admission figures, inpatients days, and ALOS. For ease of comparability over time, data for the Neonatal ward and Nursery have been combined for these figures. Occupancy rates do not include Nursery data as Nursery cradles are not considered operational beds.

**Clinical quality.**   We assessed clinical quality through two directly observed measures: crash cart stock completeness and time to triage. Crash cart inventory was captured in the Accidents & Emergency (A&E) Department, Adult Medical, and Adult Surgical wards against a 2012-established checklist. Time to triage was captured in A&E Department via a data collector recording each patient's arrival time and time taken into the triage room. Observations occurred in the morning, afternoon, and evening across multiple weekdays and a weekend day.

**Patient outcomes.**   Outcome measures included mortality by ward, pneumonia deaths in children ($\leq$14 years of age), and neonatal mortality. Deaths are shown as a percent of admissions and are stratified by those that occurred within 24 hours of admission.

Neonatal mortality was measured by dividing the number of neonates ($\leq$28 days of age) who died in the NICU by the total number of neonates admitted to the NICU. Birthweight was disaggregated into very low ($\leq$1500g), low (1501g-2499g), and normal ($\geq$2,500g) [20] Birthweight and vital status were obtained through a random sample records review at each timepoint. Between Timepoints 1 and 2, QMMH added a Neonatal ward. For 2018 data, we disaggregated data by admitting ward (NICU vs. Neonatal ward) as some neonates first admitted to the Neonatal ward died in the NICU.

### Data analysis and statistical methods

Data were entered into Microsoft® Excel and findings were compared to the Timepoint 1 results [12]. Findings are presented as the relative change per indicator. To assess statistical

significance of differences between timepoints, we performed chi-square tests of independence and Fischer's exact tests, where appropriate, in SAS version 9.4 (SAS Institute, Cary, NC), using an alpha of ≤0.05. As we did not have access to patient-level data to compare distributions around the mean for length of stay, p-values for ALOS were not calculated.

## Patient and public involvement

Patients were not directly involved in the design and conduct of the study.

## Ethics

Ethical approval was granted by the Boston University Medical Campus Institutional Review Board (Protocol H-39448) and the MoH Research and Ethics Committee in Lesotho. Permission for access to the data was also granted by Netcare Hospital Group, the primary operator of QMMH-IN. Aggregate data (at the ward and clinic levels) were received from hospital administrators and used for this analysis. Neonatal indicators required some patient-level data to understand patient outcomes based on birthweight. Researchers only received fully anonymized data. Informed consent was not possible nor required by the institutional review boards. Additional information regarding the ethical, cultural, and scientific considerations specific to inclusivity in global research is included in the (S1 Checklist).

## Results

### Capacity & utilization

Hospital beds increased between 2012 and 2018 from 390 to 410 (5.1%); filter clinic beds remained constant at 24 (Table 1). Increases in the Short Stat Medical/Surgical (35.0%), Orthopedic (16.7%), and Gynecology (40.0%) wards offset substantial decreases in the Female Surgical (-51.4%), and Pediatric Surgical (-41.2%) wards (Table 2).

Though not statistically significant, the number of clinical staff increased by 3.4%, driven heavily by physicians, increasing from 70 to 85 (p = 0.23; Table 1). Registered nurses comprised 50% of clinical staff at both timepoints. Non-clinical staff significantly decreased by 56 (17.6%, p<0.02).

Inpatient admissions decreased by 2,176 (-8.0%, p<0.0001; Table 2). The largest decreases occurred in the Female Surgical (-52.1%, p<0.0001), Pediatric Surgical (-33.3%, p<0.0001), and Adult Medical wards (Male: -34.1%, p<0.0001; Female: -27.8%, p<0.0001) as well as the filter clinics (-25.7%, p<0.0001; Table 1). The decrease in NICU admissions (-67.9%) was offset by the addition of a Neonatal ward. Admissions to the Neonatal/Nursery increased by 72.4% between 2012 (Nursery = 789) and 2018 (Neonatal = 1,243; Nursery = 117). The ICU (23.1%, p = 0.01), Step Down (25.0%, p<0.0001), and Maternity (5.7%, p = 0.0021) wards experienced the largest increases in admissions (Table 2).

Outpatient ambulatory visits increased by over 14,000 visits (3.8%, p<0.0001; Table 1), primarily at the hospital specialty outpatient clinics (25.6%, p<0.0001). Increases in visits at the A&E Department (7.0%, p<0.0001) and filter clinics (4.1%, p<0.0001) were largely offset by substantial decreases in Gateway Clinic visits (-36.8%, p<0.0001).

Hospital inpatient days increased by 21.2%, due to increased ALOS from 5.1 to 6.5 days (27.5%). ALOS increased in all wards except Short Stay Medical/Surgical (-12.2%) and ICU (-12.1%; Table 2). The longer hospital ALOS observed in 2018 was largely driven by the addition of the long-stay Neonatal ward which treated 5.5% of all hospital inpatients in 2018 but accounted for 17.3% of all hospital days. Excluding the long-stay NICU and Neonatal wards, 2018 ALOS was 5.7 days compared to 5.0 in 2012.

**Table 1. Capacity and Utilization Measures at Queen 'Mamohato Memorial Hospital Integrated Network Managed by a Public-Private Partnership in 2012 and 2018.**

| Measure [a] | 2012 | 2018 | Relative Percent Difference | p-value |
|---|---|---|---|---|
| Capacity | | | | |
| Operational beds in network [b] | 414 | 434 | 4.8% | 0.49 |
| Operational beds in hospital [b] | 390 | 410 | 5.1% | 0.48 |
| Operational beds in filter clinics | 24 | 24 | 0% | - |
| Staff members in network | 882 | 845 | -4.4% | 0.37 |
| Clinical staff members | 563 | 582 | 3.4% | 0.57 |
| Registered nurses | 284 | 295 | 3.9% | 0.65 |
| Physicians | 70 | 85 | 21.4% | 0.23 |
| Other clinical staff members | 209 | 202 | -3.3% | 0.73 |
| Non-clinical staff members | 319 | 263 | -17.6% | 0.02 |
| **Utilization** | | | | |
| Inpatient admissions [c] | 27,089 | 24,913 | -8.0% | <0.0001 |
| Hospital admissions [c,d] | 24,130 | 22,715 | -5.9% | <0.0001 |
| Filter clinic admissions [e] | 2,959 | 2,198 | -25.7% | <0.0001 |
| Inpatient days [c] | 122,656 | 151,882 | 23.8% | <0.0001 |
| Hospital days [c] | 122,656 | 148,713 | 21.2% | <0.0001 |
| Filter clinic days [e] | - | 3,169 | - | - |
| Average length of stay (days) | | | | |
| Hospital stay [c] | 5.1 | 6.5 | 27.5% | -[i] |
| Hospital stay excluding long-stay wards [c,f] | 5.0 | 5.7 | 14.0% | -[i] |
| Bed occupancy (hospital only) [g] | 82% | 99% | 21.3% | <0.0001 |
| Ambulatory care visits | 374,669 | 389,005 | 3.8% | <0.0001 |
| Hospital specialty outpatient clinic visits | 80,565 | 101,268 | 25.6% | <0.0001 |
| A&E Department visits | 20,563 | 21,993 | 7.0% | <0.0001 |
| Gateway Clinic visits | 45,733 | 28,908 | -36.8% | <0.0001 |
| Filter clinic visits | 227,605 | 236,836 | 4.1% | <0.0001 |
| % A&E visits [h] | 5.5% | 5.7% | 3.6% | 0.0016 |

Abbreviations: A&E = Accidents & Emergency.

[a] See S2 Table for detailed indicator definitions, data sources, and a description of how each indicator was constructed.

[b] Mortuary beds and nursery cradles were excluded for 2012 and 2018 figures. 2018 bed figures are an average of beds over calendar year 2018.

[c] Figures not previously published for 2012.

[d] Hospital figures for 2012 and 2018 include Nursery admissions. Only ill neonates were admitted to the nursery for observation; healthy neonates are not counted as separate admissions.

[e] Gateway Clinic does not contribute to inpatient admissions or inpatient days as it does not have beds and does not conduct deliveries.

[f] Wards with ALOS over 10 days were considered "long-stay wards" and were excluded from the sub-analysis. This included the NICU for both timepoints and the Neonatal ward for 2018 data only.

[g] Occupancy rates do not include Nursery inpatient days (numerator) or available bed days (denominator) as nursery cradles are not counted as operational beds; they do not have available bed days.

[h] % A&E visits = total visits to Accidents & Emergency Department divided by total ambulatory care visits.

[i] P-values could not be calculated for ALOS as we did not have access to patient-level data to compare distributions around the mean for length of stay.

Hospital bed occupancy increased from 82% to 99% in 2018 (19.5%, p<0.0001; Table 1), with more than half of the wards (n = 8) having occupancy rates of 90% or more (Table 2). The Neonatal ward had 209% occupancy; when beds were unavailable, neonates were shifted between the wards catering to neonates (NICU, Neonatal ward, and Nursery). Gynecology (126%) and Maternity (107%) wards also had greater than 100% occupancy.

**Table 2. Ward-level Utilization Measures at Queen 'Mamohato Memorial Hospital Integrated Network Managed by a Public-Private Partnership in 2012 and 2018.**

| Ward | Beds [a,b] | | | Admissions [a] | | | | Average Length of Stay [a,i] | | | Bed Occupancy [a,e] | | | |
|---|---|---|---|---|---|---|---|---|---|---|---|---|---|---|
| | 2012 | 2018 [b] | Relative Percent Difference | 2012 | 2018 | Relative Percent Difference | p-value | 2012 | 2018 | Relative Percent Difference | 2012 | 2018 | Relative Percent Difference | p-value |
| Short Stay Medical/ Surgical [c] | 20 | 27 | 35.0% | 1,030 | 953 | -7.5% | 0.08 | 4.1 | 3.6 | -12.2% | 58% | 36% | -37.9% | <0.0001 |
| Orthopedic | 30 | 35 | 16.7% | 1,630 | 1,461 | -10.4% | 0.0024 | 6.1 | 7.3 | 19.7% | 90% | 83% | -7.8% | <0.0001 |
| Female Medical | 30 | 29 | -3.3% | 1,865 | 1,346 | -27.8% | <0.0001 | 6.0 | 7.1 | 18.3% | 103% | 90% | -12.6% | <0.0001 |
| Male Medical | 30 | 29 | -3.3% | 1,544 | 1,017 | -34.1% | <0.0001 | 6.8 | 7.9 | 16.2% | 96% | 76% | -20.8% | <0.0001 |
| Female Surgical | 35 | 17 | -51.4% | 1,531 | 733 | -52.1% | <0.0001 | 5.7 | 7.4 | 29.8% | 69% | 90% | 30.4% | <0.0001 |
| Male Surgical | 35 | 35 | 0% | 1,953 | 1,873 | -4.1% | 0.20 | 6.8 | 6.8 | 0.0% | 103% | 99% | -3.9% | <0.0001 |
| ICU | 10 | 10 | 0% | 294 | 362 | 23.1% | 0.01 | 6.6 | 5.8 | -12.1% | 53% | 57% | 7.5% | 0.0004 |
| Gynecology | 20 | 28 | 40.0% | 2,687 | 2,322 | -13.6% | <0.0001 | 3.4 | 5.6 | 64.7% | 123% | 126% | 2.4% | 0.96 |
| Maternity [d] | 70 | 70 | 0% | 5,982 | 6,323 | 5.7% | 0.0021 | 3.5 | 4.3 | 22.9% | 81% | 107% | 32.1% | <0.0001 |
| Neonatal/ Nursery [e] | - | 33 | - | 789 | 1,360 | 72.4% | <0.0001 | 7.6 | 19.7 | 159.2% | - [e] | 209% | - | - |
| NICU [f] | 5 | 5 | 0% | 246 | 79 | -67.9% | <0.0001 | 13.5 | 20.4 | 51.1% | 181% | 88% | -51.4% | <0.0001 |
| Pediatric Medical | 31 | 31 | 0% | 1,455 | 1,208 | -17.0% | <0.0001 | 7.2 | 8.3 | 15.3% | 84% | 89% | 5.7% | <0.0001 |
| Pediatric Surgical | 34 | 20 | -41.2% | 1,492 | 995 | -33.3% | <0.0001 | 5.4 | 6.0 | 11.1% | 71% | 82% | 15.3% | <0.0001 |
| Step Down | 30 | 31 | 3.3% | 1,633 | 2,041 | 25.0% | <0.0001 | 3.3 | 5.3 | 60.6% | 49% | 94% | 91.8% | <0.0001 |
| A&E Observation [g] | 10 | 10 | 0% | - | 642 | - | - | - | 1.8 | - | - | 32% | - | - |
| Hospital Total [h] | 390 | 410 | 5.1% | 24,130 | 22,715 | -5.9% | <0.0001 | 5.1 | 6.5 | 27.5% | 82% | 99% | 20.3% | <0.0001 |

Abbreviations: ICU = Intensive Care Unit; NICU = Neonatal Intensive Care Unit; A&E = Accidents & Emergency Department.

[a] See S2 Table for detailed indicator definitions, data sources, and a description of how each indicator was constructed.

[b] Mortuary beds and nursery cradles were excluded for 2012 and 2018 figures as these are not considered operational beds. 2018 bed figures represent the average number of beds in each ward over calendar year 2018.

[c] 2012 figures are for the Ophthalmology ward. 2018 figures are for the combined Short Stay Medical/Surgical and Ophthalmology ward. See S1 Table for more information.

[d] The Maternity ward is a combination of the Antenatal and Postnatal wards for each timepoint. See S1 Table for more information.

[e] No Neonatal ward existed at QMMH in 2012 so admission and ALOS figures are for the Nursery only. For 2018, Neonatal ward and Nursery figures have been combined for ease of comparability. Only Neonatal ward cradles are considered operational beds so they are presented in the bed figures. Nursery figures have been excluded from bed occupancy figures for both timepoints as Nursery cradles are not considered operational beds.

[f] In 2018, 79 neonates were admitted directly to the NICU, though 148 neonates were transferred from the Neonatal ward to the NICU. Only direct NICU admissions were included as inpatient admissions while inpatient days, ALOS, and ward occupancy incorporated the inpatient days for any neonate who spent time in the NICU.

[g] A&E Observation treats and observes patients who were admitted through the A&E Department while they await the opening of a bed in another ward. In 2018, 642 patients were admitted to A&E Observation. 2012 data did not note A&E Observation admissions, inpatient days, or deaths.

[h] Admission and ALOS figures not previously published for 2012; we have included 2012 Nursery figures for this analysis.

[i] P-values could not be calculated as we did not have access to patient-level data to compare distributions around the mean for length of stay.

## Clinical quality

Crash carts were better equipped during Timepoint 1 observations (82.9%) compared to Timepoint 2 (73.8%; Table 3). Among missing stock during Timepoint 2, no carts stocked Heparin or the required amounts of Dopamine, Dobutamine, or Ringers Lactate. Four carts were missing electrocardiogram (ECG) leads.

During Timepoint 1, 75 patients were observed for time to triage over a total of 9 hours; during Timepoint 2, 29 patients were observed over a total of 14 hours. The proportion of

**Table 3. Clinical Quality and Patient Outcome Measures at Queen 'Mamohato Memorial Hospital Integrated Network Managed by a Public-Private Partnership in 2012 and 2018.**

| Measure [a] | 2012 | 2018 | Relative Percent Difference | p-value |
|---|---|---|---|---|
| Clinical Quality [a] | | | | |
| Stock present on crash carts [b] | 85.6% | 73.8% | -13.8% | <0.0001 |
| Patients triaged within 5 minutes in A&E [b] | 84.0% | 27.6% | -67.1% | <0.0001 |
| **Patient Outcomes** | | | | |
| Hospital mortality | 8.0% | 6.5% | -18.8% | <0.0001 |
| Mortality excluding ICU & NICU | 7.1% | 5.4% | -23.9% | <0.0001 |
| Mortality within 24 hours of admission | 28.9% | 25.9% | -10.4% | 0.0536 |
| Pediatric mortality due to pneumonia | 11.9% | 6.0% | -49.6% | 0.0083 |
| Neonatal mortality (overall) [c] | 18.0% | 16.3% | -9.1% | 0.28 |
| NICU mortality (among very low birthweight) [d] | 30.2% | 36.8% | 21.9% | 0.50 |

Abbreviations: A&E = Accidents & Emergency Department; ICU = Intensive Care Unit; NICU = Neonatal Intensive Care Unit.

[a] See S2 Table for detailed indicator definitions, data sources, and a description of how each indicator was constructed.

[b] Data for all clinical quality indicators presented were collected through direct observation in March 2013 and February 2020.

[c] Overall neonatal mortality figures are calculated using data from the NICU, Neonatal ward, and Nursery.

[d] NICU mortality among very low birthweight neonates required review of patient charts. See S2 Table for more information on methods used to conduct 2012 and 2018 chart reviews. See Fig 1 for more information on NICU mortality among very low birthweight neonates in 2018.

patients triaged within five minutes of arrival dropped from 84.0% at Timepoint 1 to 27.6% at Timepoint 2. Average time to triage observed at Timepoint 2 was 15 minutes; this was not calculated at Timepoint 1.

## Patient outcomes

Overall hospital mortality decreased from 8.0% to 6.5% (-18.8%, p<0.0001; Table 3); no deaths were reported in the filter clinics. When excluding the adult and neonatal ICUs, which treat the most critical patients and may be expected to have higher deaths, hospital mortality decreased from 7.1% to 5.4% (-23.9%, p<0.0001). Mortality rates increased significantly in the Female Surgical ward from 7.6% to 12.8% (68.4%, p<0.0001; Table 4). Mortality decreased substantially in the Nursery (-81.0%, p = 0.0006), shifting the location of death to the new Neonatal ward (13.7%).

While the number of pediatric patients admitted to the hospital with pneumonia as their primary diagnosis increased by 22.7% (2012: n = 286; 2018: n = 351; Table 3), the mortality rate nearly halved from 11.9% to 6.0% (-49.6%, p = 0.0083). Though not statistically significant, NICU mortality increased between the timepoints overall (31.7% to 36.1%, p = 0.31) and specifically among very low birthweight neonates (30.2% to 36.8%, p = 0.50). In 2018, very low birthweight neonates admitted first to the Neonatal ward had the highest mortality (48.9%; Fig 1). However, neonatal mortality overall (including all wards serving neonates; Table 3) decreased by 9.1% (p = 0.28).

## Discussion

We examined one of the first and largest healthcare PPPs in sub-Saharan Africa to understand if the performance achievements observed in the first full year of QMMH-IN operations persisted approximately six years later. We analyzed key indicators of capacity, utilization, quality, and patient outcomes and found evidence that some continued to improve while others worsened.

**Table 4. Ward-level Patient Deaths at Queen 'Mamohato Memorial Hospital Integrated Network Managed by a Public-Private Partnership in 2012 and 2018.**

| Ward | 2012 | | 2018 | | Relative Percent Difference | p-value |
|---|---|---|---|---|---|---|
| | Number of deaths | % Deaths [a] | Number of deaths | % Deaths [a] | | |
| Short Stay Medical/Surgical [b] | 9 | 0.9% | 1 | 0.1% | -88.9% | 0.02 |
| Orthopedic | 22 | 1.3% | 30 | 2.1% | 61.5% | 0.13 |
| Female Medical | 560 | 30.0% | 335 | 24.9% | -17.0% | 0.0014 |
| Male Medical | 516 | 33.4% | 290 | 28.5% | -14.7% | 0.01 |
| Female Surgical | 116 | 7.6% | 94 | 12.8% | 68.4% | <0.0001 |
| Male Surgical | 122 | 6.2% | 134 | 7.2% | 16.1% | 0.26 |
| ICU | 168 | 57.1% | 197 | 54.4% | -4.7% | 0.48 |
| Gynecology | 60 | 2.2% | 34 | 1.5% | -31.8% | 0.04 |
| Maternity [c] | 19 | 0.3% | 7 | 0.1% | -66.7% | 0.02 |
| Neonatal/Nursery [d,e] | 108 | 13.7% | 153 | 11.3% | -17.5% | 0.09 |
| NICU [e] | 78 | 31.7% | 82 | 36.1% | 13.9% | 0.31 |
| Pediatric Medical | 121 | 8.3% | 101 | 8.4% | 1.2% | 0.97 |
| Pediatric Surgical | 22 | 1.5% | 1 | 0.1% | -93.3% | 0.0005 |
| Step Down | 2 | 0.1% | 0 | 0% | -1.0% | 0.11 |
| A&E Observation [f] | - | - | 7 | 1.1% | - | - |
| **Hospital Total** | **1,923** | **8.0%** | **1,466** | **6.5%** | **-18.8%** | **<0.0001** |
| Hospital total excluding ICU & NICU | 1,677 | 7.1% | 1,269 | 6.2% | -12.7% | 0.0003 |

Abbreviations: ICU = Intensive Care Unit; NICU = Neonatal Intensive Care Unit; A&E = Accidents & Emergency Department.

[a] % Deaths = deaths per ward divided by total admissions to ward. Deaths were assigned to the ward they occurred in. A person is only admitted once to their initial ward, transfers are not included in the denominator, with minor exceptions explained below.

[b] 2012 figures are for the Ophthalmology ward. 2018 figures are for the combined Short Stay Medical/Surgical and Ophthalmology ward. See S1 Table for more information.

[c] The Maternity ward was a combination of the Antenatal and Postnatal wards for each timepoint. See S1 Table for more information.

[d] No Neonatal ward existed at QMMH in 2012, so mortality occurred only in the Nursery. For 2018, Nursery and Neonatal ward figures were combined.

[e] For 2018 data, the 148 neonates transferred from the Neonatal ward to the NICU were included in the denominator of the NICU (n = 227) to calculate the mortality rate and subtracted from the denominator of the Neonatal/Nursery (n = 1,360). If only deaths and admissions among neonates admitted directly to the NICU are considered, the NICU mortality rate for 2018 would be 35.4%.

[f] A&E Observation treats and observes patients who were admitted through the A&E Department while they await the opening of a bed in another ward. In 2018, 642 patients were admitted to A&E Observation; 7 died. 2012 data did not note A&E Observation admissions, inpatient days, or deaths.

Though hospital bed capacity increased and admissions decreased between the timepoints, bed occupancy rates increased significantly overall and in most wards. Half of the wards had occupancies at or above 90% in 2018, with 5 wards over 100% occupancy. Hospitals with bed occupancy rates above 85% have been associated with increased mortality compared to lower occupancy hospitals [21,22]. The combination of increased occupancy and limited increases in clinical staff may have overstretched resources, potentially compromising QMMH's ability to respond to infectious outbreaks or emergencies [22], a relevant concern given the current COVID-19 pandemic.

High occupancy and outpatient demand have been concerns since QMMH first opened. This has strained relations between the government and the private partner overpayment for services provided in over contract maximums [15,19]. Substantial utilization could be due to insufficient capacity or the perceived or actual reduced scope of services provided at district hospitals. Assessment of the bed and clinical service capacity at the district level, reviewing of referral practices could help redirect non-critical patients to more appropriate levels and decongest QMMH's wards.

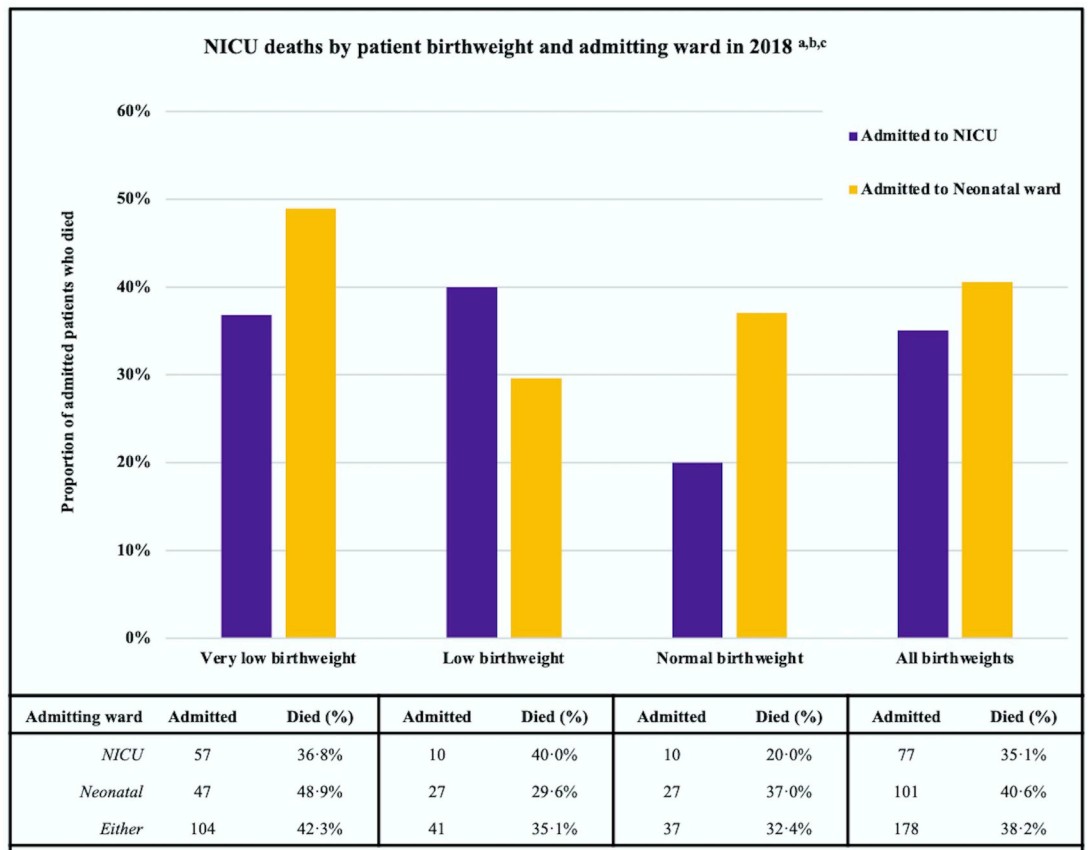

**Fig 1. Neonatal intensive care unit (NICU) deaths by birthweight and admitting ward in 2018.**

Neonatal mortality in sub-Saharan Africa is among the highest in the world, with one recent meta-analysis reporting an incidence density rate of neonatal mortality of 24.5 in NICU wards compared to 1.2 within the larger population [23]. The addition of a NICU, first brought to Lesotho through the PPP, was important and likely drove initial increases in neonatal admissions (246 in 2012). A less intensive Neonatal ward was then opened and experienced a surge in admissions (1,439 combined NICU, Neonatal ward, and Nursery admissions in 2018). While neonatal mortality overall decreased between the timepoints, NICU mortality rate at QMMH remains far above those achieved in most high-income countries, though similar to other LMICs including Uganda, India, Iran, and Nepal [24]. Mortality reviews may help

determine causes for increasing NICU death rates and should include a focus on gestational age, an important factor affecting mortality rates [24]. The high Neonatal ward utilization in 2018 (209%) suggests that population need for neonatal care far outpaces QMMH capacity; 33 incubators appear insufficient. Though not highly comparable, data from the United States suggests NICU bed rates of approximately 7.2 NICU beds per 100 births in the population, or 83–94 per 100 births under 1500 grams, are needed [25]. The need in Lesotho is likely even higher given high rates of maternal and neonatal risk factors associated with neonatal mortality including inadequate ANC care and high rates of low or very low birthweight infants [23,26]. Neonatal disorders remain the third leading cause of disability adjusted life years (DALYs) in Lesotho, driven by preterm births [27]. Antenatal, intrapartum, and postnatal interventions could reduce preterm births, neonatal trauma, and illness, decreasing the country's demand for specialty neonatal health services [28]. A systems-level approach to treat uncomplicated neonates at district hospitals and reduce demand overall for specialty neonatal health services may be needed.

Similar to experiences of other PPP hospitals such as in Brazil and Iran, where ALOS reduced by 0.6 days each to 4.8 and 4.5 days, respectively, the transition from public to PPP-governance initially resulted in a decrease in ALOS [12]. Though ALOS has increased since 2012, an ALOS of 6.5 is acceptable, particularly for a hospital meant to take the most critical cases [29,30]. QMMH's ALOS matches South Africa's 2017 country-level ALOS and is under the average of 7.7 among 36 countries in the OECD [2]. The NICU and Neonatal wards had particularly long ALOS as they treat neonates who undergo the most intensive treatments. The remaining increases in general ALOS (from 5.1 in 2012 to 5.7 in 2018 after excluding the long-stay wards) may reflect a shift in focus to treating more complicated patients as QMMH down-referred uncomplicated cases to district hospitals, though inefficient discharge procedures may also contribute.

Improvements in specific quality measurements at the PPP hospital were major findings of the 2012 study; however, our results show these started to slide in key areas. In 2020, when the direct observations took place, crash carts were less well stocked and there were longer waits for triage in the A&E Department. Though there are variety of factors that affect patient outcomes, these reduced quality measures do not appear to have affected hospital mortality rates. Quality improvement processes that focus on availability of essential emergency drugs and equipment, patient triage and patient flow, staffing, and continuous training on emergency care should be explored by hospital administrators.

Overall and in many wards, the improvements in mortality are encouraging, particularly in the ICU where ALOS also decreased. The adult medical wards accounted for 56% of hospital deaths in 2012 and 43% in 2018; focusing on these two high-volume wards may further reduce mortality. The continued improvement in the proportion of children dying of pneumonia (from 11.9% to 6.0%), might be associated with improved management of pneumonia and other infections.

Overall, QMMH-IN's performance continued halfway through the PPP contract, though with some concerning backward slides. The contentious context over the preceding years, with increasingly strained partner relationships and unpredictable cash flow, could have hampered QMMH's operations [16]. In early 2021, the GoL announced the termination of the PPP contract and intended transition of QMMH-IN to GoL management [16]. While recent events indicate the partnership itself is in serious jeopardy, our evaluation suggests that generally the hospital under the PPP was providing high-level, critically needed services in the country, and continued to improve patient outcomes. Quality at QMMH remains substantially higher than when the GoL-operated QEII experienced hospital mortality rates of 12.0%, had no triage system, and had only one crash cart [12]. The strategy of utilizing a PPP to operate the only referral hospital in Lesotho improved the range and quality of services available within the country.

Strong concerns remain over the cost of the PPP [15]. Though beyond the scope of this paper, a cost-effectiveness analysis would be beneficial to determine the cost per outcome of deaths and disability adjusted life years averted by operating QMMH-IN under the PPP. Resolution of disagreements over payment delays, extra service rates and payments, and potentially renegotiation of contract maximums would have been important to secure the partnership for the remainder of the contract and stabilize hospital performance.

## Limitations

While this is one of the first longitudinal assessments of a large-scale healthcare PPP in an LMIC, this study has several limitations. First, we did not have access to patient-level clinical data, limiting our understanding of patient mix and its contributions to ALOS or patient outcomes. Second, Timepoint 2 included administrative data from 2018 and observations in 2020. This could have resulted in some inconsistencies, making it difficult to attribute causes of changes in clinical quality. Third, our two indicators of crash cart stock and timely triage are limited measurements of the much broader and complex concept of clinical quality and cannot be generalized to the wider clinical quality of all services provided by the hospital network. They should be interpreted cautiously. The sample of patients observed for timely triage was also small overall in 2020 and compared to 2012, though the amount of observation time was greater. Extrapolating such a small sample to general operations of the A&E Department should again be done cautiously. Fourth, data do not include the birth locations of neonates treated in the NICU to understand if this impacts their survival prospects. Additionally, we have included neonates weighing exactly 1500 grams in the very low birthweight category to be consistent with 2012 data, which is slightly inconsistent with the international definition of <1500 grams [24]. Lastly, literature on NICU capacity, utilization, and mortality rates in LMICs is scarce, making it challenging to present our neonatal data within a wider context.

## Conclusion

Healthcare PPPs may be a promising mechanism to finance healthcare systems in LMICs. This study has added to scarce evidence on longitudinal performance. Within the context of a strained partnership, QMMH-IN has continued to operate, providing secondary and tertiary-level services to the country, and continuing to improve patient outcomes. Since QMMH operates like other public facilities, low-cost, high-quality, specialized medical care is, in principle, available to all Lesotho residents. This is a critical dimension of achieving universal health coverage.

## Supporting information

**S1 Checklist. Inclusivity in global research.**
(DOCX)

**S1 Table. Description of 2018 QMMH inpatient wards and changes since 2012.**
(DOCX)

**S2 Table. Indicator definitions and construction at timepoints 1 and 2.**
(DOCX)

## Acknowledgments

We would like to thank the administrators and staff at QMMH for providing access to patient records, administrative data, wards, and patients for this evaluation. We appreciate the

Ministry of Health directors and officers for supporting our data collection efforts. We thank the Lesotho-Boston Health Alliance staff in Maseru, Lesotho for their assistance and support. We would like to acknowledge our colleagues at the International Finance Corporation and the World Bank Group for providing introductions and background information.

## Author Contributions

**Conceptualization:** Nancy A. Scott, Taryn Vian.

**Data curation:** Jeanette L. Kaiser, Tshema Nash.

**Formal analysis:** Jeanette L. Kaiser, Allison Juntunen.

**Funding acquisition:** Nancy A. Scott.

**Investigation:** Nancy A. Scott, Taryn Vian.

**Methodology:** Nancy A. Scott, Brian W. Jack, Taryn Vian.

**Project administration:** Nancy A. Scott, Jeanette L. Kaiser, Elizabeth L. Nkabane–Nkholongo.

**Supervision:** Nancy A. Scott, Taryn Vian.

**Validation:** Nancy A. Scott, Brian W. Jack, Taryn Vian.

**Visualization:** Jeanette L. Kaiser, Brian W. Jack, Taryn Vian.

**Writing – original draft:** Nancy A. Scott, Jeanette L. Kaiser.

**Writing – review & editing:** Nancy A. Scott, Jeanette L. Kaiser, Brian W. Jack, Elizabeth L. Nkabane–Nkholongo, Allison Juntunen, Tshema Nash, Mayowa Alade, Taryn Vian.

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
