## [Decision Letter · Decision Letter 0]

7 Mar 2022

PONE-D-21-38208Observational study of the clinical performance of a Public-Private Partnership national referral hospital network in Lesotho: Do improvements last over time?PLOS ONE

Dear Dr. Scott,

Thank you for submitting your manuscript to PLOS ONE. After careful consideration, we feel that it has merit but does not fully meet PLOS ONE’s publication criteria as it currently stands. Therefore, we invite you to submit a revised version of the manuscript that addresses the points raised during the review process.

We look forward to receiving your revised manuscript.

Kind regards,

Paavani Atluri

Academic Editor

PLOS ONE

Journal Requirements:

3. Please include a complete copy of PLOS’ questionnaire on inclusivity in global research in your revised manuscript. Our policy for research in this area aims to improve transparency in the reporting of research performed outside of researchers’ own country or community. The policy applies to researchers who have travelled to a different country to conduct research, research with Indigenous populations or their lands, and research on cultural artefacts. The questionnaire can also be requested at the journal’s discretion for any other submissions, even if these conditions are not met.  Please find more information on the policy and a link to download a blank copy of the questionnaire here: https://journals.plos.org/plosone/s/best-practices-in-research-reporting. Please upload a completed version of your questionnaire as Supporting Information when you resubmit your manuscript.

Reviewers' comments:

Reviewer's Responses to Questions

**Comments to the Author**

1. Is the manuscript technically sound, and do the data support the conclusions?

Reviewer #1: No

Reviewer #2: Yes

2. Has the statistical analysis been performed appropriately and rigorously? 

Reviewer #1: I Don't Know

Reviewer #2: Yes

3. Have the authors made all data underlying the findings in their manuscript fully available?

Reviewer #1: Yes

Reviewer #2: Yes

4. Is the manuscript presented in an intelligible fashion and written in standard English?

Reviewer #1: Yes

Reviewer #2: Yes

5. Review Comments to the Author

Reviewer #1: This study examined a few metrics at a referral hospital in Lesotho, to ascertain if improvements seen at inception are sustained, in the context of a strained corporate relationship. It found that capacity and utilisation of the hospital increased, whilst patient outcomes improved in most areas, and quality decreased. These findings were interpreted as a promising sign of the financing model employed by this hospital, which has been, to date, poorly characterised in lower-middle income countries.

In general, this is a well-written paper. It requires a minimal language review (e.g. the abbreviation DALYs needs to be explained), and perhaps a clearer description of the aim of the paper in the abstract.

The study setting requires some clarity – does QMMH function as a combined district/regional/tertiary hospital? Where does Gateway clinic refer a patient if district-level hospital services are indicated?

The metrics selected are subject to many factors that are not really discussed, which is why I answered "No" to "Is the manuscript technically sound, and do the data support the conclusions?" Some examples inlcude a perverse incentive to increase bed utilisation rate in light of private funding, competency of individual staff/a team, competency of clinical/corporate leaders, patients’ perceptions of quality, different staff being hired in different areas – ophthalmology beds increasing due to greater number of ophthalmologists?? etc. The PPP may have driven some of these changes, but how did it drive these? The data analysed is not granular enough to reveal this. Without acknowledgement of this, I see this study as having a questionable “valid contribution to the base of academic knowledge” (as per email from PLOS One).

Quality measures are poorly selected; in addition, the number of patients assessed for time to triage is very small.

A revision of the neonatal results and discussion is required. At this stage, it has the potential to be confusing. The description of the Nursery is explained in S1 – I feel it would be wise to include this explanation in the main body of the text, so that a reader clearly understands from an early stage the differences between NICU, Neonatal ward and Nursery. Also hidden in the supplemental information (S2) is whether Nursery beds are included in the network’s operational beds – please clarify in the main body of the text. I would not exclude “admissions” to nursery from the capacity/utilisation data, when deaths (i.e. outcomes that are later analysed) occur in this area. I suggest revising how neonatal mortality is calculated using neonates admitted to NICU as the denominator: what about the neonates who were admitted to the Neonatal ward and then died? Disaggregation by the admitting ward confuses things; quality of care (and outcome) is not dependent on the ward a patient was admitted.

Regarding statistics, I am unsure about the use of the mean square; besides being unfamiliar with it as a statistical test, and whether it was appropriately employed, I am also unsure what the results mean. The authors present the mean squares of ALoS in Table 2, but then don’t really comment on what their significance.

It is mentioned in lines 173-6 that “We conducted simple linear regression models with a dichotomous variable indicating the year to assess differences between ALOS for the timepoints. As we did not have access to patient-level data, p-values were not calculated. We present the mean square for each regression model.”, but then contradict this in lines 211/212 “Hospital inpatient days increased by 29.0%, due to increased ALOS from 5.0 to 6.5 days (30.9%, p=0.08)”. i.e. p-valued is calculated and presented.

The discussion overall is long. It does a reasonable job of condensing the large volume of data analysed, but it should be more concise, in terms of sticking to answering the research question. When it comes to discussion of the neonatal results, some of the literature quoted is not naturally comparable to the data presented.

In the Limitations section, there is too much commentary around the birthweight categories. This can simply be acknowledged.

Reviewer #2: This is a great article discussing the impact of public private partnership in a LMIC. I think it is well written with ample statistical data. I have the following comments with some minor edits.

The authors chose to use crash cart supplies as a surrogate for quality, however the reason of how this relates to quality is not well discussed? Also was this an impact of lack of funding is not known. Usually low supplies can be extrapolated to higher mortality and lower quality. It would be helpful to the explain this earlier. How does a lower supplies explain a higher quality or lower mortality rate?

The data clearly shows some of the pitfalls of public private partnerships. The reason for higher ophthalmology beds with lower occupancy is certainly an outlier that does not make sense when the overall occupancy rate is higher. Was there a directive for the partnership that allots increased ophthalmology care (for e.g. a government drive to eradicate cataracts by wider screening and surgery for older patients).

The facts that this hospital is heavily directed for maternal and neonatal care should also be highlighted. Though these usually translate into higher neonatal and infant mortality, was there a difference in overall mortality reduction from cardiovascular or oncologic mortality is not known given the limited medical admissions or beds. Did this have any impact on quality?

Also it would be helpful to know the scope of the public private partnership. What is the monetary impact of a per percentage drop in mortality. Did every 0.1% drop in mortality cost %100,000 or $1million is not known. What was the funding commitment or resource reimbursement per case is a data point that will help to tag a cost to the quality achieved and help benchmark future funds. Was this data available over the time period of 2012 to 2018

6. PLOS authors have the option to publish the peer review history of their article (what does this mean?). If published, this will include your full peer review and any attached files.

Reviewer #1: **Yes: **Adam Konrad Asghar

Reviewer #2: **Yes: **Bright Thilagar

---

## [Author Response · Author response to Decision Letter 0]

7 May 2022

We thank the reviewers for their comments, suggestions, and critiques. We have addressed each point

they brought up below.

---

## [Decision Letter · Decision Letter 1]

15 Jun 2022

PONE-D-21-38208R1Observational study of the clinical performance of a Public-Private Partnership national referral hospital network in Lesotho: Do improvements last over time?PLOS ONE

Dear Dr. Scott,

Thank you for submitting your manuscript to PLOS ONE. After careful consideration, we feel that it has merit but does not fully meet PLOS ONE’s publication criteria as it currently stands. Therefore, we invite you to submit a revised version of the manuscript that addresses the points raised during the review process.

We look forward to receiving your revised manuscript.

Kind regards,

Paavani Atluri

Academic Editor

PLOS ONE

Journal Requirements:

Reviewers' comments:

Reviewer's Responses to Questions

**Comments to the Author**

1. If the authors have adequately addressed your comments raised in a previous round of review and you feel that this manuscript is now acceptable for publication, you may indicate that here to bypass the “Comments to the Author” section, enter your conflict of interest statement in the “Confidential to Editor” section, and submit your "Accept" recommendation.

Reviewer #1: All comments have been addressed

2. Is the manuscript technically sound, and do the data support the conclusions?

Reviewer #1: Yes

3. Has the statistical analysis been performed appropriately and rigorously? 

Reviewer #1: Yes

4. Have the authors made all data underlying the findings in their manuscript fully available?

Reviewer #1: Yes

5. Is the manuscript presented in an intelligible fashion and written in standard English?

Reviewer #1: Yes

6. Review Comments to the Author

Reviewer #1: Thank you for addressing all of my comments, and for engaging in a robust discussion around the points - it has been educational for me!

The main reason I have selected "Minor Revision" is because the results in the abstract need correcting in light of the updated results (utilisation/ALoS).

I would like to take the opportunity though, to suggest considering inclusion of some of your rebuttal in the Discussion - you pose very valuable points in response to my critique. The reviewer/reader may fault the metrics/indicators, but ultimately the aim of the research was to describe changes in these over time, rather than thoroughly explore or validate them. As I say, this is just a consideration... however, I do feel it would contribute to the quality of the manuscript.

e.g.

"The indicators selected were appropriate at the time of the baseline assessment in 2009. Because it was not initially designed as a longitudinal study and because of the evolving context of the PPP and partners, some of the changes in indicators may have been driven by multiple factors."

"...we opted to use the same measures that were initially selected in part to ensure comparability over time and ensure the utility of findings for key stakeholders including the MoH and the World Bank."

7. PLOS authors have the option to publish the peer review history of their article (what does this mean?). If published, this will include your full peer review and any attached files.

Reviewer #1: **Yes: **Adam Konrad Asghar

---

## [Author Response · Author response to Decision Letter 1]

29 Jun 2022

1. The main reason I have selected "Minor Revision" is because the results in the abstract need correcting in light of the updated results (utilisation/ALoS).

a. Response: Thank you for catching this. We have updated the abstract accordingly. We have also updated the abstract and main text to reflect that the PPP has formally dissolved, seven years earlier than anticipated. 

2. I would like to take the opportunity though, to suggest considering inclusion of some of your rebuttal in the Discussion - you pose very valuable points in response to my critique. The reviewer/reader may fault the metrics/indicators, but ultimately the aim of the research was to describe changes in these over time, rather than thoroughly explore or validate them. As I say, this is just a consideration... however, I do feel it would contribute to the quality of the manuscript. e.g. "The indicators selected were appropriate at the time of the baseline assessment in 2009. Because it was not initially designed as a longitudinal study and because of the evolving context of the PPP and partners, some of the changes in indicators may have been driven by multiple factors."

"...we opted to use the same measures that were initially selected in part to ensure comparability over time and ensure the utility of findings for key stakeholders including the MoH and the World Bank."

a. Response: Thank you for this suggestion. We’ve included what you’ve suggested in the methods section: “The indicators selected were appropriate at the time of the baseline assessment in 2009. Because this was not initially designed as a longitudinal study and because of the evolving context of the PPP, some indicators changed over time and other measures of quality were not included. We opted to use the same measures that were initially selected to ensure comparability over time and utility for key stakeholders.

---

## [Editor Report · Decision Letter 2]

22 Jul 2022

Observational study of the clinical performance of a Public-Private Partnership national referral hospital network in Lesotho: Do improvements last over time?

PONE-D-21-38208R2

Dear Dr. Scott,

We’re pleased to inform you that your manuscript has been judged scientifically suitable for publication and will be formally accepted for publication once it meets all outstanding technical requirements.

Kind regards,

Paavani Atluri

Academic Editor

PLOS ONE
---

## [Editor Report · Acceptance letter]

2 Sep 2022

PONE-D-21-38208R2 

Observational study of the clinical performance of a Public-Private Partnership national referral hospital network in Lesotho: Do improvements last over time? 

Dear Dr. Scott:

I'm pleased to inform you that your manuscript has been deemed suitable for publication in PLOS ONE. Congratulations! Your manuscript is now with our production department. 

Kind regards, 

on behalf of

Dr. Paavani Atluri 

Academic Editor

PLOS ONE